# Extracellular Release of Mitochondrial DNA: Triggered by Cigarette Smoke and Detected in COPD

**DOI:** 10.3390/cells11030369

**Published:** 2022-01-22

**Authors:** Luca Giordano, Alyssa D. Gregory, Mireia Pérez Verdaguer, Sarah A. Ware, Hayley Harvey, Evan DeVallance, Tomasz Brzoska, Prithu Sundd, Yingze Zhang, Frank C. Sciurba, Steven D. Shapiro, Brett A. Kaufman

**Affiliations:** 1Center for Metabolism and Mitochondrial Medicine, Division of Cardiology, Department of Medicine, University of Pittsburgh, Pittsburgh, PA 15261, USA; sad115@pitt.edu (S.A.W.); hah85@pitt.edu (H.H.); 2Heart, Lung, and Blood Vascular Medicine Institute, University of Pittsburgh, Pittsburgh, PA 15261, USA; devallan@pitt.edu (E.D.); brzoskat@pitt.edu (T.B.); prs51@pitt.edu (P.S.); 3Division of Pulmonary, Allergy and Critical Care Medicine, Department of Medicine, University of Pittsburgh, Pittsburgh, PA 15213, USA; adg56@pitt.edu (A.D.G.); zhang3@pitt.edu (Y.Z.); fcs@pitt.edu (F.C.S.); shapirosd@upmc.edu (S.D.S.); 4Department of Cell Biology, School of Medicine, University of Pittsburgh, Pittsburgh, PA 15261, USA; mip85@pitt.edu; 5Division of Hematology/Oncology, School of Medicine, University of Pittsburgh, Pittsburgh, PA 15261, USA; 6Department of Bioengineering, University of Pittsburgh, Pittsburgh, PA 15261, USA

**Keywords:** cigarette smoke, COPD, mitochondria, cell-free DNA, oxidative stress, senescence, necroptosis, extracellular vesicles

## Abstract

Cigarette smoke (CS) is the most common risk factor for chronic obstructive pulmonary disease (COPD). The present study aimed to elucidate whether mtDNA is released upon CS exposure and is detected in the plasma of former smokers affected by COPD as a possible consequence of airway damage. We measured cell-free mtDNA (cf-mtDNA) and nuclear DNA (cf-nDNA) in COPD patient plasma and mouse serum with CS-induced emphysema. The plasma of patients with COPD and serum of mice with CS-induced emphysema showed increased cf-mtDNA levels. In cell culture, exposure to a sublethal dose of CSE decreased mitochondrial membrane potential, increased oxidative stress, dysregulated mitochondrial dynamics, and triggered mtDNA release in extracellular vesicles (EVs). Mitochondrial DNA release into EVs occurred concomitantly with increased expression of markers that associate with DNA damage responses, including DNase III, DNA-sensing receptors (cGAS and NLRP3), proinflammatory cytokines (IL-1β, IL-6, IL-8, IL-18, and CXCL2), and markers of senescence (p16 and p21); the majority of the responses are also triggered by cytosolic DNA delivery *in vitro*. Exposure to a lethal CSE dose preferentially induced mtDNA and nDNA release in the cell debris. Collectively, the results of this study associate markers of mitochondrial stress, inflammation, and senescence with mtDNA release induced by CSE exposure. Because high cf-mtDNA is detected in the plasma of COPD patients and serum of mice with emphysema, our findings support the future study of cf-mtDNA as a marker of mitochondrial stress in response to CS exposure and COPD pathology.

## 1. Introduction

Chronic obstructive pulmonary disease (COPD) is the third leading cause of death worldwide. It accounts for around three million deaths per year, affects 250 million people worldwide, and is accompanied by social and economic burdens [1]. COPD is defined by progressive airflow limitation due to dysregulated chronic inflammation that obstructs the small airways (bronchiolitis) or destroys the lung parenchyma (emphysema). Structural remodeling of airway wall thickness and pulmonary vasculature compromise oxygen exchange, and can lead to long-term disability and early death [2].

COPD is caused by genetic mutations, aberrant cellular responses to bacterial and viral infections, or chronic exposure to indoor and outdoor air pollutants, allergens, chemical toxins, and cigarette smoke (CS)—the latter being the most common risk factor [2]. Current therapies include bronchodilators, corticosteroids, and antibiotics to slow disease progression, but no existing medications prevent long-term lung decline. Furthermore, the mechanisms that drive the inflammation and subsequent tissue remodeling in COPD are not completely understood. Notably, a typical feature of chronic inflammation in COPD-affected lungs is the failure to recover even after several years of smoking cessation, suggesting that autoimmunity is a significant driver of the ongoing inflammatory process [3,4].

Several studies have indicated that mitochondrial dysfunction is a hallmark of COPD [5,6,7,8,9,10]. CS contains two potent inhibitors of mitochondrial respiratory complex IV, i.e., carbon monoxide and hydrogen cyanide [11]. Furthermore, CS has been shown to inhibit electron transfer from mitochondrial complexes I, II, and III [6,9]. The inhibition of the respiratory chain promotes electron leakage and consequently generates superoxide anion, which is derived into additional reactive oxygen species (ROS) [12]. Bioenergetic impairment (decreased ATP production), oxidative stress, and accumulation of dysmorphic mitochondria are common features of cells exposed to CS [5,6] and of epithelial cells from lung sections of COPD patients [7,8]. Furthermore, CS-induced lung dysfunction and tissue damage are attenuated by mitochondrial intervention strategies (i) to reduce ROS production by mitochondrial iron chelation or bypassing mitochondrial complex III/IV inhibition [8,9] or (ii) to replace dysfunctional mitochondria by mitochondrial transplantation [10].

Of note, mitochondria are the only extranuclear organelles in animal eukaryotic cells that maintain a genome (mitochondrial DNA, mtDNA). Stress can trigger mtDNA release from the mitochondria to the cytosol or extracellular space (released also into the blood) [13]. Outside of the mitochondria, mtDNA is considered to be a damage-associated molecular pattern molecule for its ability to trigger the innate immune system by engaging multiple DNA-sensing receptors (DSRs) and activating downstream proinflammatory signaling cascades. A subset of interactions with major DSRs have been described for mtDNA signaling: (i) cyclic GMP-AMP synthase (cGAS) recognizes naked mtDNA or mtDNA packaged with mitochondrial transcription factor A (TFAM) [14,15]; (ii) nucleotide-binding domain-like receptor protein 3 (NLRP3) binds oxidized mtDNA [16]; and (iii) toll-like receptor 9 (TLR9) recognizes hypomethylated CpG mtDNA sequences [17]. Furthermore, toll-like receptor 4 (TLR4) activation during cell injury induces mtDNA fragmentation by endonuclease G, and consequently, contributes to sustained inflammation [18].

Mitochondrial DNA damage may be a significant contributor to its mislocalization, and is a major target of benzo(α)pyrene (and its derivatives), which are found in CS [19]. In addition, mtDNA is more exposed to oxidative damage because it is more loosely packaged than nuclear DNA (nDNA) and is located near a superoxide source (complexes I and III) [20]. Alveolar macrophages collected from bronchoalveolar lavage (BAL) from current smokers showed two-fold more nDNA and six-fold more mtDNA damage than non-smokers, as assessed by the percentage of mtDNA common deletions, which was consistent with decreased mtDNA copy number [21]. Moreover, a small cohort of lung tissues (*n* = 4) collected from end-stage COPD (GOLD IV) patients showed a substantial increase in mtDNA strand breaks and abasic sites as compared with unaffected lungs [22]. Because mtDNA damage could be the initiating event in mtDNA release from the mitochondria, these prior reports prompted us to investigate mtDNA mislocalization in models of CS exposure and COPD pathogenesis. Here, we test the hypothesis that CS exposure promotes extracellular mtDNA release, which may also be detected in former smokers with COPD as a consequence of airway damage. We found increased levels of cf-mtDNA in the plasma of former smokers with COPD and the serum of mice with CS-induced emphysema. As direct evidence, we found mtDNA release by extracellular vesicles (EVs) and cell debris in human bronchial epithelial cells exposed to sublethal doses of cigarette smoke extract (CSE).

## 2. Materials and Methods

### 2.1. Human Lung Parenchyma and Plasma Collection

Lung tissues were obtained from 12 COPD patients after lung transplantation at the University of Pittsburgh Medical Center. Normal lung tissues were obtained from 12 donated organs not suitable for transplantation through the Center for Organ Recovery and Education (CORE). Plasma was used from 14 former smokers without lung disease and 20 former smokers diagnosed with COPD randomly selected among the participants from the University of Pittsburgh Specialized Center for Clinically Oriented Research (SCCOR) [23]. All available demographics and lung physiology for human plasma and lung tissue cohorts are described in the Appendix A. The COPD lung tissue collection (STUDY18100070) and the SCCOR study (STUDY19090239) were approved by the Institutional Review Board for Human Subject Research at the University of Pittsburgh.

### 2.2. Animal Experiments

Animal studies were performed in accordance with the Institutional Animal Care and Use Committee of the University of Pittsburgh (protocol #18113916). Using a five-chamber smoking apparatus to deliver smoke to mice that were maintained in a monitored, flow-regulated fume hood [24], ten-week-old female mice were exposed to CS from four unfiltered 3R4F reference cigarettes (College of Agriculture, University of Kentucky, Lexington, KY) per day, five days per week, for six months. Control female mice were exposed to room air. Mice were euthanized by CO_2_ inhalation, followed by immediate cardiac puncture to collect the blood, and then were tracheostomized. The right lobe of the lung was dissected and snap-frozen for protein extraction. The left lobe was inflated with 10% formalin at constant pressure for 10 min (min), fixed, and paraffin embedded. Blood was allowed to clot in an untreated tube for one hour (h) at room temperature and centrifuged at 11,200× *g* for 10 min at 4 °C. Serum (100 μL) was then collected and stored at −80 °C.

### 2.3. Cigarette Smoke Extract (CSE) Preparation

CSE was prepared by burning one 3R4F reference cigarette and bubbling the direct and side-stream smoke into a 50 mL conical tube containing 25 mL of DMEM/F12 (Gibco, cat. #11320-033) without FBS and antibiotics using a peristaltic pump (Precision Blood Pump, COBE Perfusion System) at 0.12 rotations per minute (rpm). The CSE-medium was filtered through a 0.22 μm syringe filter (Fisher Scientific, Waltham, MA, USA, cat# 09720004), and the optical density (OD) was measured at a wavelength of 310 nm using a Synergy H1 (BioTek, Winooski, VT, USA) microplate reader. These solutions had OD_310_ = 0.578 ± 0.016, were defined as 100% CSE, and then diluted to obtain the desired concentrations (i.e., doses). Fresh CSE was prepared for each experiment and used within one h of preparation.

### 2.4. Statistical Analysis

The statistical analysis was performed using GraphPad Prism (version 9.1, GraphPad Software, Inc., San Diego, California, USA). Data are presented as mean ± SEM or as median and quartiles (violin or box and whisker plots). After verifying the normality, differences between the two groups were analyzed by Student’s *t*-test, whereas differences among more than two groups were analyzed by one-way or two-way ANOVA with indicated *post hoc* analyses. Statistical significance was set at * *p* < 0.05, ** *p* < 0.01, *** *p* < 0.001, and **** *p* < 0.0001.

Additional methods are reported in the Appendix A.

## 3. Results

### 3.1. Cell Free-mtDNA and -nDNA Are Higher in Human Plasma from Former Smokers with COPD Than Former Smokers without COPD

Because COPD is characterized by extensive lung damage and inflammation [4] and mislocalized (i.e., cytosolic and extracellular) mtDNA could contribute to inflammation, we determined whether the altered abundance of cell-free DNA could be associated with COPD. We measured cell-free mtDNA (cf-mtDNA) and nuclear DNA (cf-nDNA) in the plasma of former smokers with COPD and former smokers without any clinical evidence of obstruction (Appendix A). The levels of both cf-mtDNA and cf-nDNA were elevated in former smokers with COPD as compared with the unobstructed former smoker group (Figure 1A–B). In a randomly selected subgroup of the tested plasma, long-range polymerase chain reaction (PCR) of five overlapping amplicons (encompassing the complete 16.6 kb circular mitochondrial genome) suggested the presence of intact or large fragments of the mitochondrial genome (Appendix A).

Necroptosis has been recently emphasized among several maladaptive pathways involved in COPD pathogenesis [7,24], including senescence [25], autophagy [26,27], and apoptosis [27]. Necroptosis is a caspase-independent cell death process that promotes a controlled cell membrane lysis and may facilitate DNA release into the extracellular space [28]. From lung tissue collected from COPD patients or donors without signs of obstruction (Appendix A), we measured protein levels of the upstream regulators of necroptosis, receptor-interacting serine/threonine protein kinase 1 and 3 (RIPK1 and RIPK3, respectively). We did not observe a significant change in RIPK1 between the two groups. However, we measured a two-fold increase in the level of RIPK3 in the COPD group as compared with the control (Appendix A). Our findings suggest that the upregulation of necroptosis marker RIPK3 is associated with mtDNA and nDNA release observed in COPD patients.

### 3.2. Cell Free-mtDNA Levels Are Increased in the Serum of Mice with Alveolar Destruction Induced by Cigarette Smoke

To evaluate cf-mtDNA levels in an animal model, we exposed mice to CS for six months and quantified the lung damage by measuring the mean alveolar chord length. As expected, we found enlarged alveolar airspace (emphysema) in CS-exposed mice as compared with the room air-exposed group (Figure 1C,D). In the same cohort of mice, we measured cf-mtDNA and cf-nDNA levels in the serum. CS-exposed mice showed almost three-fold higher cf-mtDNA levels than those exposed to room air (Figure 1E). The cf-nDNA levels followed a similar trend but were not statistically significant (Figure 1F). The necroptotic RIPK1 and RIPK3 proteins were higher in the lung lysates of CS-exposed mice than room air controls (Appendix A). These results indicate that, in a mouse model with emphysema, CS exposure upregulates necroptosis markers and is associated with cf-mtDNA release.

### 3.3. Cigarette Smoke Extract Inhibits Cell Proliferation and Promotes Cell Death in Bronchial Epithelial Cells

To identify the molecular mechanisms underlying our in vivo observations (Figure 1E,F), we employed a common *in vitro* cell model to study CS effects [7,8,24]. Specifically, human bronchial epithelial cells (BEAS-2B) were exposed to increasing doses of CSE for 24 h, and their phenotype was characterized. A sulforhodamine B assay revealed that CSE toxicity was dose dependent (Figure 2A). Increased cell death was observed at 20% and 30% CSE, evidenced by the rounded and shrinking cell morphology. Cells incubated with 10% CSE were spared this severe effect with only a 22% decrease in cell density as compared with unexposed cells (Figure 2A,B). To test whether cell death was related to the genotoxic effect of CSE, we measured *de novo* genomic DNA synthesis by 5-ethynyl-2′-deoxyuridine (EdU) incorporation during S-phase. DNA synthesis was almost absent (75.8% decline relative to unexposed) in cells exposed to 20% CSE, whereas cells exposed to 10% CSE showed only a 17.5% decline (Figure 2C). Because DNA double-strand breaks (DSB) induce permanent growth arrest and cell death, we measured the number of p53-binding protein 1 (53BP1) foci, a protein involved in DSB signaling and repair [29]. Cells exposed to 20% CSE showed more than double the foci per nucleus than the control (Figure 2D,E). On the contrary, 10% CSE promoted selective recruitment of 53BP1, as indicated by the increased volume of the foci (Figure 2D,F) without increasing foci number (Figure 2E). These observations indicate that exposure to 20% CSE mainly triggers cell death (lethal), whereas a lower dose (10%) affects cell proliferation by slowing genomic DNA synthesis (sublethal).

### 3.4. Mitochondrial DNA Is Released into the Extracellular Milieu via Vesicles following CSE Exposure

To test whether DNA is released into the extracellular milieu of bronchial epithelial cells, we exposed cells to increasing doses of CSE for 24 h. From the growth medium, we collected EVs and cell debris. A nanoparticle tracking analysis of the EVs showed no variation in the abundance or size of the particles among the control, 5, and 10% CSE preparations. In contrast, the EVs from 20% CSE preparations were significantly more abundant (1.9 × 10^9^ *versus* 9.31 × 10^8^ particles per mL) and slightly larger than the control (150.5 nm *versus* 120.5 nm in diameter) (Figure 3A, Appendix A). In the EVs, mtDNA content increased with the CSE dose, whereas no statistically significant variation of nDNA abundance was detected as compared with the control (Figure 3B,C). Long-range PCR of the DNA isolated from the EVs showed the presence of the whole or large fragments of the mtDNA genome (Appendix A), in line with the previous results in human plasma (Appendix A). The mtDNA and nDNA contents both increased in the cell debris for cells exposed to CSE (5–20%) as compared with unexposed cells (Figure 3E,F). Notably, the ratio of mtDNA to nDNA in the EVs increased in a CSE dose-dependent manner (Figure 3D), while the ratio decreased in the cell debris (Figure 3G). These results suggest that exposure to moderate (5–10%) doses of CSE promotes mtDNA release by living cells via EVs. On the contrary, both mtDNA and nDNA are massively released by cell debris due to cell death caused by CSE cytotoxicity. We speculate that the relative genome levels in these compartments represent distinctive cellular responses.

### 3.5. CSE Promotes Mitochondrial Membrane Depolarization, Superoxide Production, and Oxidative Stress

Because mtDNA release could be driven by perturbations of mitochondrial homeostasis [13,30], we evaluated the effects of CSE on mitochondrial membrane potential and superoxide anion production as direct indicators of mitochondrial function. We found that CSE promoted mitochondrial membrane depolarization (Figure 4A) and increased superoxide production (Figure 4B) in a dose-dependent manner. Pretreatment with the mitochondrial scavenger MitoTEMPO prevented superoxide overproduction, confirming that the measurements were specific for mitochondrial superoxide (Appendix A).

Although moderate levels of superoxide-derived ROS play a role in cellular signaling and adaptation, high ROS levels can lead to protein oxidation and degradation [31]. To evaluate the long-term effects of superoxide overproduction, we first quantified carbonylated protein levels, a major product of ROS-mediated oxidation reactions. We found that total carbonylated proteins were modestly increased at 10% CSE, but widely detected at 20% CSE (Figure 4C,D). Next, we measured the total ubiquitinated proteins as a marker of protein degradation and the levels of the adaptor protein sequestosome 1 (SQSTM1, i.e., p62), which recruits ubiquitinated proteins and organelles to the autophagosome [32]. Levels of p62 are used to monitor autophagic flux, with elevated levels indicating that flux is inhibited [32]. Similar to the results of carbonylation, total ubiquitinated and p62 protein levels increased in cells exposed to 10% CSE and increased further with 20% CSE exposure (Figure 4E–G). These data indicate that CSE exposure induces mitochondrial membrane depolarization, increases superoxide production, causes protein oxidation, and reduces autophagic flux even at sublethal doses (i.e., 10% CSE).

### 3.6. Exposure to a Sublethal Dose of CSE Promotes Cellular Senescence

The mechanisms of mtDNA release remain poorly understood. Specific cell death pathways (necrosis and necroptosis) have been proposed to be involved in mtDNA release following severe cell damage [13,30]. Cellular stress induced by CS and CSE has been shown to promote mitophagy-dependent necroptosis [7] and replicative senescence [33]. To test whether increased necroptosis was associated with the process of mtDNA release in cells exposed to a sublethal dose of CSE (10%), we investigated the levels of RIPK1, RIPK3, and mixed lineage kinase domain-like (MLKL, downstream of RIPK1 and RIPK3 in the necroptotic cascade) markers. Both RIPK1 and MLKL proteins showed no substantial changes in abundance in cells exposed to CSE as compared with the control (Appendix A). RIPK3 decreased proportionally to the CSE dose (Appendix A) rather than increased as observed in the lung of COPD patients and mice exposed to CS (Appendix A).

Because we observed nDNA synthesis inhibition in cells exposed to CSE (Figure 2C), we next tested whether senescence markers were activated. First, we analyzed cell senescence by β-galactosidase (β-gal) staining. Cells exposed to 5–20% CSE showed an increased number of β-galactosidase positive cells as compared with unexposed cells (Figure 4H,I), with 10% showing peak levels. To confirm replicative senescence in the cells exposed to 10% CSE, we measured the proteins levels of the cyclin-dependent kinase inhibitor 1A (p21) as a marker of cell cycle arrest. Western blotting showed a 2.8-fold increase in the level of p21 in cells exposed to 10% CSE as compared with the control (Figure 4J,K); p21 and cyclin-dependent kinase inhibitor 2A (p16) mRNAs were both also increased (Figure 4L), demonstrating a transcriptional response to block cell cycle progression in cells exposed to 10% CSE. These results suggest that exposure to a sublethal dose of CSE induces mtDNA release in EVs and promotes cellular senescence without activation of necroptosis.

### 3.7. Mitochondrial Dynamics Are Altered in Cells Exposed to a Sublethal Dose of CSE and in COPD Lung Tissue

Perturbations to mitochondrial dynamics may also favor mtDNA release [13]. To test whether mitochondrial structure was altered, we performed a morphometric analysis, using translocase of the outer mitochondrial membrane 20 (TOMM20) immunostaining to visualize the mitochondrial network. Morphometry showed that the number of mitochondria per cell did not change at 10% CSE (Figure 5A,B), in agreement with the observation that intracellular mtDNA content (mtDNA/nDNA) and TFAM (whose levels usually correlate with mtDNA abundance) were also unchanged (Appendix A). However, the total mitochondrial area and perimeter per cell, and the mean area, perimeter, aspect ratio, and form factor per mitochondrion were increased (Figure 5A,C–H). These results indicate that exposure to a sublethal dose of CSE promotes mitochondrial enlargement and elongation.

To enhance our understanding of this process, we quantified protein markers of mitochondrial fusion, including the longform of optic atrophy 1 (L-OPA1) and mitofusin-1 and -2 (MFN1 and MFN2), which are involved in tethering the inner and outer mitochondrial membranes [34]. L-OPA1, MFN1, and MFN2 showed a slight but significant decrease (29% for L-OPA, 20% for MFN1 and MFN2) in cells exposed to 10% CSE (Figure 5I,J,L,M), whereas no changes were observed for the short form of OPA1 (S-OPA1, Figure 5K). Similarly, MFN1 and MFN2 proteins were decreased in COPD lung samples as compared with the lung tissue of donors without signs of obstruction, confirming dysregulated mitochondrial dynamics in the pathological tissues (Figure 5N–P). Consistent with the literature [5,35,36], these findings show markers of altered mitochondrial dynamics in cells exposed to CS and in COPD lung tissue. The altered mitochondrial dynamics associated with cf-mtDNA is potentially a common feature and may be functionally relevant to mtDNA release [13]. However, the molecular mechanisms in CSE-stimulated release require further study.

### 3.8. Exposure to a Sublethal Dose of CSE Upregulates the Expression of DNase III, cGAS, NLRP3, and TLR4

The accumulation of mislocalized DNA in our epithelial culture model could be caused by impaired DNases and could lead to the activation of DSRs [37]. In cells exposed to a sublethal dose (10%) of CSE, we quantified the expression of the main classes of DNases, differentiated by their activity and localization: DNase I (extracellular), DNase II (phagolysosomal), and DNase III (i.e., TREX1, cytosolic) [37]. While we did not observe significant differences in DNases I and II expression, DNase III showed an almost 50% increase in cells exposed to 10% CSE as compared with unexposed cells (Figure 6A), suggesting the activation of a cytosolic mechanism to digest mislocalized DNA.

To identify possible receptors for mislocalized mtDNA, we evaluated the expression of DSRs mainly localized to the cytoplasm that could trigger an inflammatory response caused by CS exposure, specifically cGAS and NLRP3. Furthermore, we measured TLR4, which is involved in mtDNA fragmentation [18], and whose deficiency causes age-dependent emphysema [38]. cGAS, NLRP3, and TLR4 expression increased in cells incubated with 10% CSE as compared with unexposed cells (Figure 6B). The increased expression of DNase III together with cGAS, NLRP3, and TLR4 suggests that DNA also accumulates in the cytoplasm and may engage DSRs to promote an inflammatory response.

### 3.9. Exposure to a Sublethal Dose of CSE Upregulates the Expression of Proinflammatory Cytokines Involved in the Recruitment of Neutrophils and Macrophages

Inflammation in the lungs of COPD patients is characterized by increased pro-inflammatory cytokines [4,39], some of which are downstream products of signaling cascades triggered by mislocalized DNA [14,15,16,17,30]. Because neutrophil migration in the airways and inflammation are prominent features of COPD [40,41], we measured the expression of IL-8, the major chemoattractant of neutrophils in the lungs [41,42], as well as Chemokine C-X-C motif 1 and 2 (CXCL1 and CXCL2, respectively). Cells exposed to 10% CSE did not show significant variation of CXCL1 expression as compared with unexposed cells, whereas IL-8 and CXCL2 were strongly upregulated (Figure 6C), implying a contribution of bronchial epithelial cells to attract neutrophils.

Subsequently, we analyzed the gene expression of IL-1β, IL-6, and IL-18. Chronic expression of IL-1β causes lung inflammation and emphysema by activating lung macrophages [4,43,44]. IL-6 plays a major role in the systemic inflammation observed in COPD patients [45], and its overexpression in murine lung cells causes airway inflammation and airspace enlargement [46]. High levels of IL-18 have been found in the lungs of COPD patients and smokers [47], and its overexpression in mouse lungs triggered inflammatory cell infiltration and alveolar enlargement [48]. In cells exposed to 10% CSE, IL-1β and IL-6 were strongly upregulated while IL-18 was increased to a lesser extent (Figure 6D). Overall, these data indicate that exposure to a sublethal CSE dose induces the transcription of cytokines and chemokines, suggesting an active role for bronchial epithelial cells in recruiting neutrophils and macrophages.

## 4. Result

### Transfection of Synthetic DNA to Mimic Delivery of cf-DNA by EVs Upregulates the Expression of DNase III, cGAS, NLRP3, TLR4, and Proinflammatory Cytokines Involved in the Recruitment of Neutrophils and Macrophages

To analyze the effect of cf-DNA on bronchial epithelial cells without the interference of other compounds contained in CSE, we treated BEAS-2B cells with a repetitive double-stranded DNA sequence of poly(dA:dT), a synthetic analog of B-DNA. We used two strategies. Cells were incubated with 1 or 2 µg/mL of synthetic DNA that was added directly to the growth medium for 12 h or were transfected using a cationic lipid-based transfection reagent. When poly(dA:dT) DNA was added directly to the growth medium without transfection reagent, we did not observe a significant increase in the gene expression of DNases I, II, III (Figure 7A–C), cGAS, NLRP3, TLR4 (Figure 7D–F), and proinflammatory cytokines (Figure 7G–L) as compared with the untreated cells (i.e., mock, extracellular). Interestingly, cells transfected with 1 or 2 µg/mL of synthetic DNA showed a substantial increase in the expression of DNase III (Figure 7C), cGAS, NLRP3, TLR4 (Figure 7D–F), IL-8, CXCL1, CXCL2 (Figure 7G–I), IL-1β, and IL-6 (Figure 7J,K) as compared with cells treated only with transfection reagent (without polydA:dT, i.e., mock, transfected), and to untreated and non-transfected cells (i.e., mock, extracellular). We did not observe significant differences in the expression of DNase I, DNase II, and IL-18 with poly(dA:dT) transfection (Figure 7A,B,L). These findings indicate that cf-DNA internalized by bronchial epithelial cells is sufficient to induce upregulation of the majority of the DNA-sensing receptors and proinflammatory cytokines involved in the recruitment of neutrophils and macrophages that are likewise upregulated by CSE exposure (Figure 6).

## 5. Discussion

COPD, which is responsible for three million deaths worldwide each year, is caused mostly by cigarette smoking and has no resolutive therapeutics [1,2]. Excessive airway inflammation and remodeling remain even after several years of smoking cessation, suggesting autoimmunity as a significant driver of the ongoing processes [3,4]. In the last decade, mislocalized mtDNA has been shown to promote inflammatory signaling [14,15,16,17]. In this study, we wanted to determine whether mtDNA was released upon CS exposure and could be detected in the plasma of former smokers with COPD due to airway damage. Measuring extracellular mtDNA and understanding its role may identify novel therapeutic targets for smokers and COPD patients.

Our data revealed elevated levels of cf-mtDNA in the plasma of former smokers with COPD (Figure 1A,B) and in the serum of mice with CS-induced enlarged airspace (Figure 1C–F, Appendix A). The elevated cf-mtDNA has precedent in other human COPD studies. In a small single-center study, higher levels of total cf-DNA (measured by spectrophotometry) were detected in the plasma of patients with COPD exacerbations admitted to hospital as compared with COPD patients without exacerbations and healthy controls, and were associated with an increased risk of 5-year mortality [49]. Recently, in a subcohort from the Subpopulation and Intermediate Outcome Measures in COPD Study (SPIROMICS), elevated plasma cf-mtDNA levels were observed in patients with mild and moderate COPD as compared with smokers without airflow obstruction [50]. Similarly, urine cf-mtDNA levels have been associated with increased respiratory symptoms among smokers, and correlated with worse spirometry and chest computed tomography scans in males with emphysema, and with worse respiratory symptoms in females [51].

To understand the mechanism of mtDNA release, we exposed BEAS-2B cells to a sublethal dose of (10%) CSE (Figure 2A,B) and observed decreased mitochondrial membrane potential, increased oxidative stress (Figure 4A–G), and altered markers of mitochondrial dynamics (Figure 5A–M). These changes occurred concomitantly with replicative senescence, as demonstrated by the expression of senescent markers and cell cycle inhibition (Figure 2C and Figure 4H–L). Noteworthy, we described two ways of mtDNA release by EVs and cell debris (Figure 3B,E). EVs showed a relative increase in mtDNA content over nDNA with increasing CSE doses, while cell debris showed a relative decrease (Figure 3D,G) (Appendix A), suggesting a different paradigm of release. The function and destination of CSE-induced EVs have not been established.

Necroptosis has been shown to be a driving mechanism of cell death caused by exposure to high doses of (16–20%) CSE and in the lung of COPD patients [7,24]. We found an upregulation of RIPK3 in COPD lungs and RIPK1 and RIPK3 in CS-exposed murine lungs (Appendix A), which suggested that necroptosis was occurring and was a possible mechanism of mtDNA release. However, RIPK1 and RIPK3 were not upregulated in cells exposed to a sublethal (10%) dose of CSE (Appendix A). This observation was in agreement with a recent study that used the same cell line, where necroptosis was not driven by upregulation of RIPK3, but instead by phosphorylation of MLKL [24]. Furthermore, in our study, FBS was not included in the media to avoid EV contamination [52,53], and its absence may have influenced cell death signaling, including necroptosis

Interestingly, BEAS-2B cells have been shown to increase exosome release when exposed to CSE. This process is driven by the thiol-reactive properties of CSE, especially acrolein. Mechanistically, CSE oxidizes the thiol groups of membrane proteins exposed to the extracellular milieu (exofacial thiols) to promote membrane fusion and subsequent exosome release [54,55]. In agreement, we found that cells exposed to a sublethal dose of CSE had increased oxidative stress and released mtDNA in EVs (Figure 3B and Figure 4A–G). Oxidative stress may also alter mitochondrial dynamics and is one of the mechanisms believed to promote mtDNA packaging into EVs, or its release in the cytosol [13]. Indeed, we showed a decrease in MFN1, MFN2, and L-OPA levels in BEAS-2B exposed to a sublethal dose of CSE and in emphysematous human lung tissues (Figure 5I,J,L–P, Appendix A).

One possible explanation for CSE-induced mitochondrial elongation (Figure 5A–H) is provided by the observation that during autophagy, some mitochondria elongate as a compensatory mechanism to produce ATP and sustain viability under stressful conditions [56]. This process could precede mitochondrial fission [57] and may lead to damaged mitochondria (including mtDNA) at the cell periphery [56] to, then, be extruded in EVs. Recently, it has been shown that when damage overwhelmed autophagy, dysfunctional mitochondria were released from cardiomyocytes to be taken up and degraded by macrophages [58]. Alternatively, because long-range PCR suggested the presence of the whole mtDNA genome in human plasma and EVs (Appendix A), it is also plausible to hypothesize an increased horizontal transfer of mtDNA or whole mitochondria between cells to compensate for the bioenergetic defects induced by CS and protect against lung injury, as observed in others models [59]. An alternative signal activating mtDNA release could be IL-1β (Figure 6D), which has been shown to induce the release of mtDNA into the cytoplasm [60]. Additional considerations are in the Appendix A.

In our *in vitro* model (Figure 8), CSE exposure induced nDNA release predominantly by cell debris (Figure 3C,F), supporting the idea that it was released by dead cells, as observed in neutrophils [61]. On the contrary, mtDNA was released by cell debris and EVs (Figure 3B,E), trended with decreased mitochondrial membrane potential, increased superoxide, and protein oxidation (Figure 4A–D). The release of mtDNA was further associated with upregulated DSRs (cGAS and NLRP3), DNA responsive genes (DNase III), and proinflammatory cytokines characteristic of COPD inflammation (Figure 6A–D). Although not proven in this case, this suggests that mislocalized mtDNA in cells exposed to a sublethal dose of CSE could contribute to activating an immune response by cGAS (IL-6), NLRP3 (IL-1β/IL-18), or TLR-9 (IL-1β/IL-6/IL-8) signaling (as observed with synthetic DNA, Figure 7D–K) [14], and may act as an autocrine and paracrine signal that precedes cell death. The presence of mislocalized DNA could also trigger compensatory expression of DNases (Figure 7C) to avoid the immune response. A recent report showed that DNase I treatment could diminish the pathogenic effects of acute CS exposure in mouse lungs [62]. This result supports the hypothesis that the extracellular release of DNA following CS exposure contributes to disease etiology. Notably, cf-mtDNA could be the primary driver of the pathophysiological role of cf-DNA induced by CS exposure and in COPD because of its high copy number (relative to nDNA) (Figure 1A,B,E,F and Figure 3B,C,E,F). While the present study has demonstrated substantial support for our hypothesis, it has some limitations (Appendix A).

Our findings support that cf-mtDNA could be a marker of mitochondrial stress in response to CS exposure and in COPD, warranting further study. Understanding the mechanism of mtDNA release, and its extracellular role may identify novel therapeutic targets for smokers and COPD patients. For instance, the use of DNases as therapeutics could alleviate the immunogenic role of cf-DNA [62]. Furthermore, cf-mtDNA levels in the plasma of COPD patients could be used to monitor systemic mitochondrial dysfunction, disease progression, and treatment efficacy (especially with treatments involving mitochondrial transfer, ROS scavengers, mitochondrial biogenesis stimulators, iron chelators, and mitophagy inhibitors) [8,9,10,59,63,64,65].

Supplementary discussions are reported in the Appendix A.

## Figures and Tables

**Figure 1 cells-11-00369-f001:**
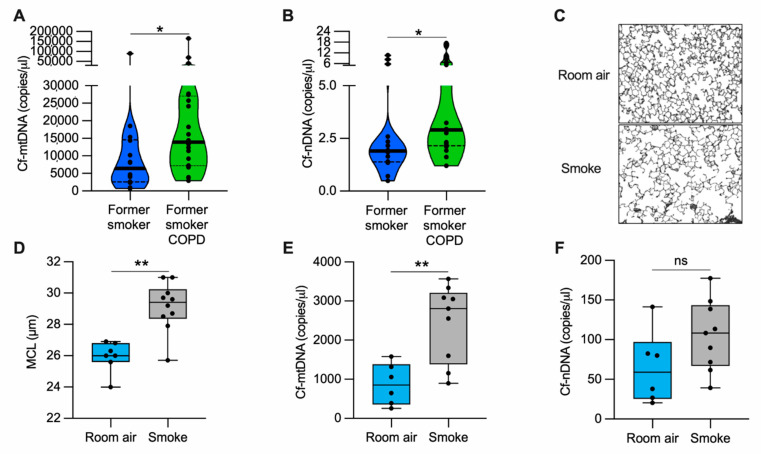
cf-mtDNA is elevated in the plasma of former smokers with COPD and in the serum of a mouse model of emphysema induced by CS exposure. (**A**) Cell-free mitochondrial DNA (cf-mtDNA) and (**B**) cell-free nuclear DNA (cf-nDNA) levels in the plasma of former smokers without airway obstruction (*n* = 14) and former smokers with COPD (*n* = 20); (**C**) representative images of modified Gill’s stained lung tissue shown in black and white as the threshold for the analysis (original magnification 200×); (**D**) mean chord length (MCL) quantification of mice exposed to cigarette smoke (*n* = 10) or room air (*n* = 7) for six months; (**E**) cf-mtDNA and (**F**) cf-nDNA in the serum of mice exposed to cigarette smoke (*n* = 9) or room air (*n* = 6) for six months. Data are shown in the (**A**,**B**) truncated violin or in the (**D**–**F**) box and whiskers plots, with circles representing individual samples. The bold horizontal bands in the (**A**,**B**) truncated violin plots represent the median (second quartile), and the dashed bands are the first and the third quartiles. ns, no statistical significance; * *p* < 0.05 and ** *p* < 0.01 were determined by an unpaired *t*-test.

**Figure 2 cells-11-00369-f002:**
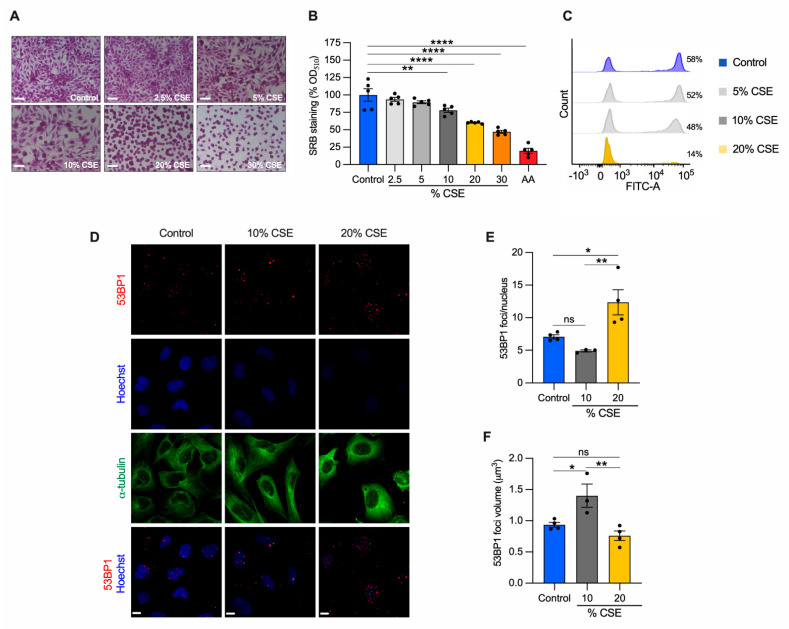
CSE inhibits cell proliferation, promotes nDNA damage, and cell death. BEAS-2B cells exposed to the indicated CSE dose (%) for 24 h were (**A**) stained with sulforhodamine B (SRB) and (**B**) quantified by measuring the absorbance at 510 nm and normalized to unexposed cells (control, shown as 100%). Antimycin A (AA) was used as a positive control for cell toxicity. Scale bar = 100 µm; (**C**) representative analysis of flow cytometry showing the number of cells (%) engaged in *de novo* DNA synthesis by EdU incorporation (*n* = 2); (**D**) representative maximum projections of confocal image stacks of 53BP1 (red), the nucleus (Hoechst, blue), and the cytoskeleton (α-tubulin, green); (**E**) the number of 53BP1 foci per nucleus, and (**F**) their volume in control, 10%, and 20% CSE-treated cells. Scale bar = 10 µm. (**B**,**E**,**F**) Data are reported as mean ± SEM of independent experiments indicated as circles (*n* ≥ 3). CSE, cigarette smoke extract; control, CSE-unexposed cells. ns, no statistical significance; * *p* < 0.05, ** *p* < 0.01, **** *p* < 0.0001 were determined by one-way ANOVA with multiple comparisons using (**B**) Dunnett’s or (**E**,**F**) Tukey’s *post hoc* tests.

**Figure 3 cells-11-00369-f003:**
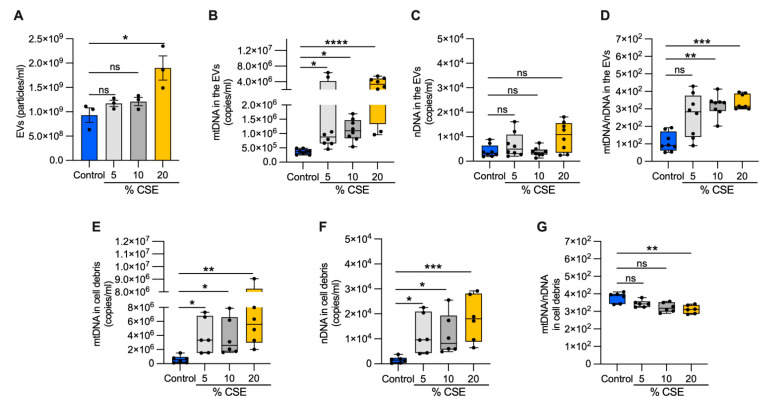
Exposure to CSE promotes mtDNA release in EVs. (**A**) Bar graph showing the EV number (particles/mL) isolated from the extracellular milieu of BEAS-2B exposed for 24 h to CSE exposure; box and whisker plots of the (**B**) mtDNA and (**C**) nDNA contents; (**D**) the mtDNA/nDNA ratio from isolated EVs as well as the (**E**) mtDNA and (**F**) nDNA content; (**G**) the mtDNA/nDNA ratio from isolated cell debris. (**A**) Data are shown as mean ± SEM of independent experiments indicated as a circle (*n* = 3). (**B**–**G**) In the box and whisker plot, each circle represents the value of an independent experiment (*n* ≥ 6). CSE, cigarette smoke extract; control, CSE-unexposed cells. (**A**–**G**) ns, no statistical significance; * *p* < 0.05, ** *p* <0.01, *** *p* < 0.001, **** *p* < 0.0001 were determined by one-way ANOVA with multiple comparisons and Kruskal–Wallis’s test relative to the control.

**Figure 4 cells-11-00369-f004:**
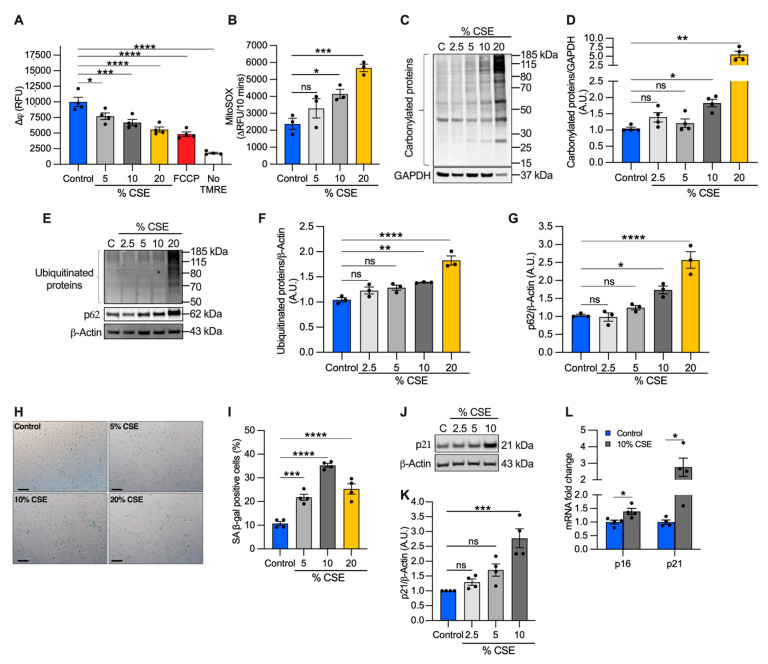
CSE affects mitochondrial membrane potential (∆***ψ***), promotes oxidative stress, and induces replicative senescence. (**A**) Bar graph reporting the ∆***ψ*** as relative fluorescence unit of BEAS-2B cells exposed for 6 h to CSE and incubated for 20 min with 250 nM tetramethylrhodamine ethyl ester (TMRE). Cells incubated without TMRE or with 20 µM carbonyl cyanide p-trifluoro-methoxyphenyl hydrazone (FCCP) were used as negative and positive controls, respectively; (**B**) bar graph reporting the superoxide anion production as the variation of the relative fluorescence unit (RFU) of MitoSOX over 10 min in BEAS-2B cells exposed for 3 h to CSE. Representative Western blots showing total (**C**) carbonylated and (**E**) ubiquitinated proteins, (**E**) p62 and (**J**) p21 proteins in BEAS-2B cells exposed for 24 h to CSE. GAPDH or β-actin proteins were used as a loading control. Bar graphs report the densitometry of total (**D**) carbonylated and (**F**) ubiquitinated proteins, and (**G**) p62 and (**K**) p21 proteins; (**H**) representative micrograph images and (**I**) quantification (percentage of total cells) of senescent-associated β-galactosidase positive cells after 24 h of exposure to CSE. Scale bar = 100 µm; (**L**) bar graph reporting the gene expression of p16 and p21 in cells exposed for 24 h to a sublethal dose (10%) of CSE as compared with the control. (**A**,**B**,**D**,**F**,**G**,**I**,**K**,**L**) Data are shown as mean ± SEM of independent experiments (*n* ≥ 3) indicated as circles. CSE, cigarette smoke extract; control, CSE-unexposed cells; ns, no statistical significance; **p* < 0.05, ** *p* < 0.01, *** *p* < 0.001, **** *p* < 0.0001, (**A**,**B**,**D**,**F**,**G**,**I**) by one-way ANOVA analysis as compared with the control with Bonferroni *post hoc* test, or (**K**) Kruskal–Wallis *post hoc* tests, and (**L**) unpaired *t*-test with Mann–Whitney analysis.

**Figure 5 cells-11-00369-f005:**
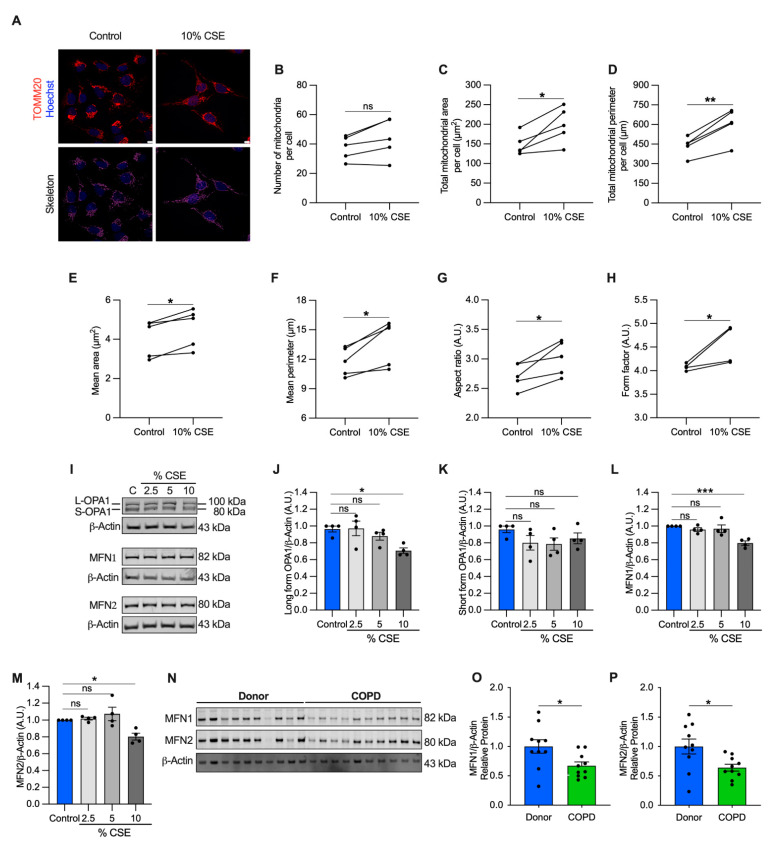
Mitochondrial dynamics are dysregulated in cells exposed to a sublethal dose of CSE and in the airways of COPD patients. (**A**) Representative maximum projections of 3D-confocal images show the mitochondrial network immunolabeled with TOMM20 (red) in control and cells exposed to 10% CSE for 24 h. Nucleus (Hoechst, blue) was used as counterstaining. Greyscale images were converted to binary to generate a skeleton (magenta). Scale bar = 10 µm; graphs represent (**B**) the number of mitochondria, total mitochondrial (**C**) area and (**D**) perimeter per cell, and the (**E**) mean area, (**F**) mean perimeter, (**G**) aspect ratio (major/minor axis length), and (**H**) form factor (inverse of sphericity) per single mitochondrion; (**I**) representative Western blot of OPA1 (long and short forms, L- and S-, respectively), MFN1, and MFN2 protein levels in BEAS-2B cells exposed for 24 h to increasing doses of (2.5, 5, and 10%) CSE. Bar graphs showing the densitometry of (**J**) long and (**K**) short forms of OPA-1, (**L**) MFN1, and (**M**) MFN2 proteins; (**N**) representative Western blot showing MFN1 and MFN2 proteins, and their relative (**O**,**P**) densitometry in the airways of human donors without signs of COPD (*n* = 10) or affected by COPD (*n* = 10). β-actin protein was used as a loading control. (**B**–**H**) Data are shown as the mean of 6–8 analyzed images per condition for each independent experiment (*n* = 5), as mean ± SEM of (**J**–**M**) independent experiments (*n* = 4), or (**O**,**P**) several human samples indicated as circles. CSE, cigarette smoke extract; control, CSE-unexposed cells. ns, no statistical significance; * *p* < 0.05, ** *p* < 0.01, *** *p* < 0.001 determined by a (**B**–**H**) two-tailed paired *t*-test, (**J**–**M**) one-way ANOVA analysis as compared with the control with Bonferroni *post hoc* test, (**O**,**P**) unpaired *t*-test analysis with Welch’s correction.

**Figure 6 cells-11-00369-f006:**
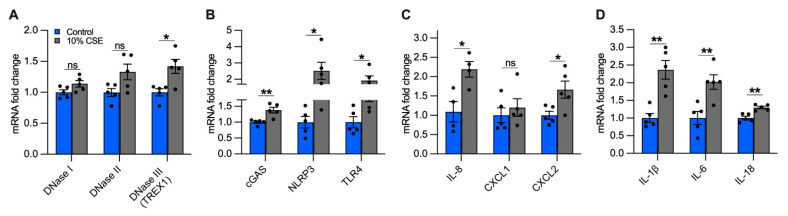
A sublethal dose of CSE upregulates the expression of DNase III, DNA-sensing receptors, and cytokines involved in neutrophil and macrophage recruitment. Bar graphs reporting the gene expression in cells exposed for 24 h to a sublethal dose of (10%) CSE as compared with the control: (**A**) DNases I-III, (**B**) cGAS, NLRP3, TLR4, (**C**) IL-8, CXCL1, CXCL2, and (**D**) IL-1β, IL-6, IL-18. Data are shown as mean ± SEM of independent experiments (*n* ≥ 4), indicated as circles. CSE, cigarette smoke extract; control, CSE-unexposed cells. ns, no statistical significance; * *p* < 0.05, ** *p* < 0.01 were determined by an unpaired *t*-test analysis with Welch’s correction.

**Figure 7 cells-11-00369-f007:**
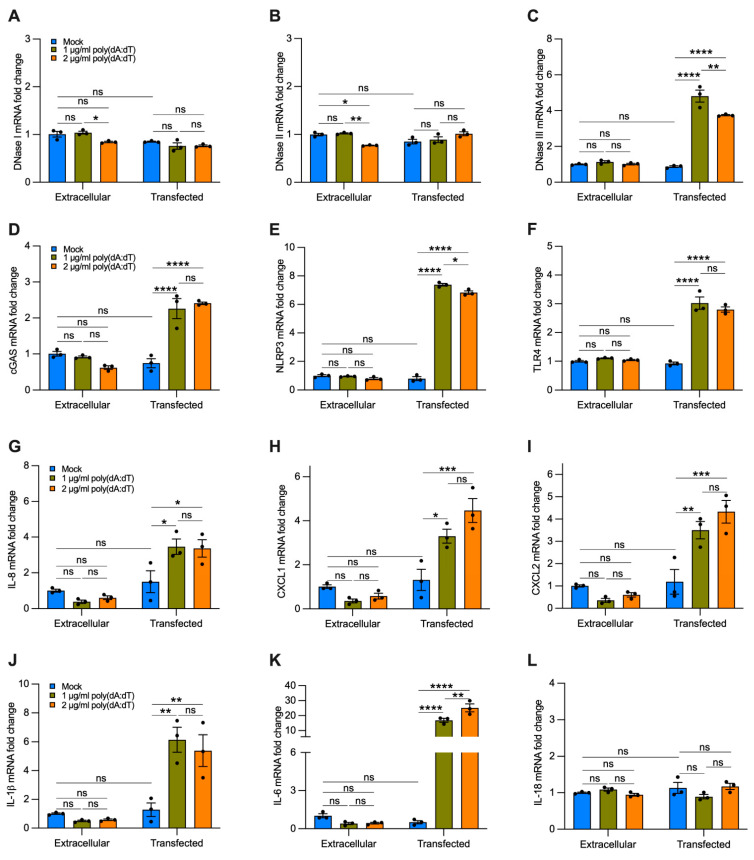
Transfected synthetic DNA upregulates the expression of DNase III, DNA-sensing receptors, and cytokines involved in neutrophil and macrophage recruitment. Bar graphs reporting the gene expression in cells exposed for 12 h to 1 or 2 µg/mL of synthetic DNA added directly to the growth medium (extracellular) or transfected using a cationic lipid-based transfection reagent (transfected): (**A**–**C**) DNases I-III; (**D**–**F**) cGAS, NLRP3, TLR4; (**G**–**I**) IL-8, CXCL1, CXCL2; (**J**–**L**) IL-1β, IL-6, IL-18. Data are shown as mean ± SEM of independent experiments (*n* = 3), with each experiment indicated as a circle. ns, no statistical significance; * *p* < 0.05, ** *p* < 0.01, *** *p* < 0.001, **** *p* < 0.0001 were determined by two-way ANOVA with multiple comparisons.

**Figure 8 cells-11-00369-f008:**
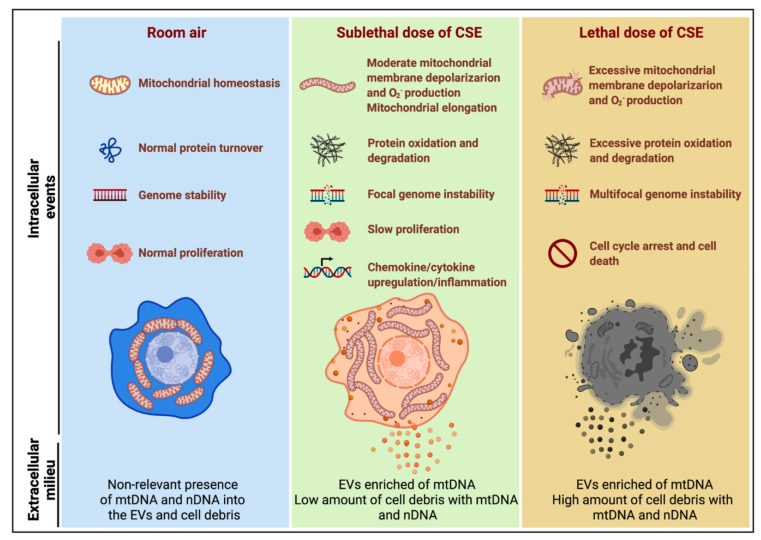
Model summarizing the effects of sublethal (10%) or lethal (20%) doses of CSE with extracellular mtDNA and nDNA release. A sublethal dose of CSE promoted moderate mitochondrial membrane depolarization, increased superoxide (O_2_^−^) production and oxidative stress, slowed cell proliferation, and triggered mtDNA release by extracellular vesicles (EVs). This condition is exacerbated in cells exposed to a lethal dose of CSE, in which mtDNA and nDNA are mainly released by the high amount of cell debris.

## Data Availability

Data are available from the corresponding authors upon reasonable request.

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
