# Peer review of "Extracellular Release of Mitochondrial DNA: Triggered by Cigarette Smoke and Detected in COPD"

_cells, 2022, doi:10.3390/cells11030369_

Round 1

Reviewer 1 Report

Summary

This is a very interesting, well written, structured and illustrated article detailing a systematic series of investigations into extracellular mtDNA (and DNA) in relation to cigarette smoke and COPD. I only have minor comments (detailed below). Conditional on these being addressed adequately, I would recommend this article for publication in the journal ‘Cells’.

Abstract

Lines 37-38: For the non-expert, if you have not done so, please elaborate briefly why you are looking at the expression of certain markers.

Materials and methods

Line 109: I can see in the SI the characteristics of the patients, but what about the tissues from the ‘normal’ individuals?

Results

Lines 162-165: Any differences in mtDNA deletions between the treatment groups?

Discussion

Lines 432-433: You say “Measuring 432 extracellular mtDNA and understanding its role may identify novel therapeutic targets 433 for smokers and COPD patients.” Towards the end of the discussion, I feel it would be appropriate for you to consider and to include a brief discussion on how indeed your findings could contribute to the development of therapeutic targets.

SI

Line 63: Use appropriate protein abbreviation for mitochondrial NADH-ubiquinone oxidoreductase core subunit 1 ie MT-ND1 rather than ND1PL

Line 188: Long range PCR of what? I can guess but I would rather you say.

Reviewer 2 Report

Line 48: ‘excluding COVID-19’. Unrelated citation. Please remove.

Lines 66-67: explain better the inhibition of respiratory complexes related to the overexpression of ROS, because respiratory complexes are the source of ROS.

Line 77: ‘including the blood circulation’. The release into the blood is a consequence, thus better to say: ‘released also into the blood’. Please, modify accordingly.

Line 175-177: the conclusion is not enough based on a simple association between a western blot and PCR. You can only conclude an association. Please, formulate again the sentence in agreement with the given comment.

Line 201: please change ‘and triggers’ with ‘is associated’.

Line 252: ‘mechanisms of DNA release and signaling’. The presence or the amount of released nDNA ans mtDNA cannot be related to mechanisms and signalling, since the sort of experiments doesn’t allow to discern about the mechanism and signalling involved. Please, revise the sentence.

Line 254: which is the evidence about the ‘programmed’ release? The term programmed is used in improper way. Please, modify and revise.

Line 290: the authors cannot conclude increased protein degradation by p62 increased level. The protein p62 covers big complexes of damaged proteins and/or damaged organelles promoting mainly the autophagy. Atrogenes are mainly involved in the vehiculation to the proteasome system, driving damaged proteins for proteasome degradation. However, even considering p62 level as indicator of degradation and autophagy: the level of p62 if augmented indicates reduced autophagic flux and\or blocked autophagy flux at some level. Thus, level of beclin and LC3 clarify the entire chain of events. The authors are invited to provide other experiments related to the autophagy process or revise the sentence in agreement with the literature on autophagy and protein degradation.

Line 357: why the authors conclude the altered dynamics can bring to mtDNA release?

Line 440: the authors state ‘altered mitochondrial dynamics’, but they have measured only biomarkers in blot and not microscopy analysis. Thus, they can only state about altered mitochondrial dynamics biomarkers suggesting altered mitochondrial dynamics.

Line 443: it’s not extrusion. The authors don’t describe the extrusion mechanism, but they only measure evidence suggesting extrusion. Why do they write about two ways? Which is the hypothesis based on literature?

Line 454: why do they avoid using FBS? Which is the literature suggesting such approach?

Other questions:

Is there any control with cells or mice exposed to the smoke machine vapour without the nicotine or the main components of the smoke?

For nuclear DNA: why it has been chosen the gene indicated as reference of nuclear DNA?

The presentation of data and some conclusions should be improved. The discussion is too long and should be highlighted which is the real aim of the authors related to the present study and the comparison among different approaches in vitro.

Round 2

Reviewer 2 Report

Thanks for the appropriate revisions. The paper has been improved.